# Let-7a Downregulation Accompanied by KRAS Mutation Is Predictive of Lung Cancer Onset in Cigarette Smoke–Exposed Mice

**DOI:** 10.3390/ijms241411778

**Published:** 2023-07-21

**Authors:** Alessandra Pulliero, Luca Mastracci, Letizia Tarantini, Zumama Khalid, Valentina Bollati, Alberto Izzotti

**Affiliations:** 1Department of Health Sciences, University of Genoa, 16132 Genoa, Italy; zumama.khalid@gmail.com; 2Department of Surgical Sciences and Integrated Diagnostics (DISC), Anatomic Pathology, University of Genoa, 16132 Genoa, Italy; luca.mastracci@unige.it; 3IRCCS Ospedale Policlinico San Martino, 16132 Genova, Italy; izzotti@unige.it; 4Epiget Lab, Department of Clinical Sciences and Community Health, University of Milan, 20122 Milan, Italy; letizia.tarantini@unimi.it (L.T.); valentina.bollati@unimi.it (V.B.); 5Fondazione IRCCS Ca’ Granda Ospedale Maggiore Policlinico, 20122 Milan, Italy; 6Department of Experimental Medicine, University of Genoa, 16132 Genoa, Italy

**Keywords:** microRNA, KRAS mutation, let-7a expression, cigarette smoke, lung cancer

## Abstract

Background: Let-7 is a tumor suppressor microRNA targeting the KRAS lung oncogene. Let-7a downregulation is reversible during the early stages of lung carcinogenesis but is irreversible in cancer cells. The aim of this study is to shed light on the relationship between oncogene (KRAS) mutation and let-7a downregulation in cigarette smoke (CS)-induced lung carcinogenesis. Methods: A total of 184 strain H Swiss albino mice were either unexposed (control) or exposed to CS for 2 weeks (short CS) or 8 months (long CS). After 8 months, the lungs were individually collected. The following end points have been evaluated: (a) DNA methylation of the let-7a gene promoter by bisulphite-PCR and pyrosequencing; (b) let-7a expression by qPCR; (c) KRAS mutation by DNA pyrosequencing; (d) cancer incidence by histopathological examination. Results: let-7a expression decreased by 8.3% in the mice exposed to CS for two weeks (CS short) and by 33.4% (*p* ≤ 0.01) in the mice exposed to CS for 8 months (CS long). No significant difference was detected in the rate of let-7a-promoter methylation between the Sham-exposed mice (55.1%) and the CS short-(53%) or CS long (51%)-exposed mice. The percentage of G/T transversions in KRAS codons 12 and 13 increased from 2.3% (Sham) to 6.4% in CS short– and to 11.5% in CS long–exposed mice. Cancer incidence increased significantly in the CS long–exposed mice (11%) as compared to both the Sham (4%) and the CS short–exposed (2%) mice. In the CS long–exposed mice, the correlation between let-7a expression and the number of KRAS mutations was positive (R = +0.5506) in the cancer-free mice and negative (R = −0.5568) in the cancer-bearing mice. Conclusions: The effects of CS-induced mutations in KRAS are neutralized by the high expression of let-7a in cancer-free mice (positive correlation) but not in cancer-bearing mice where an irreversible let-7a downregulation occurs (negative correlation). This result provides evidence that both genetic (high load of KRAS mutation) and epigenetic alterations (let-7a irreversible downregulation) are required to produce lung cancer in CS-exposed organisms.

## 1. Introduction

Lung cancer is the leading cause of cancer death worldwide, accounting for nearly 10 million deaths in 2020 [1]. Cigarette smoking is the major cause of lung cancer [2]. Tobacco use remains the leading preventable cause of death in the US, accounting for about 1 in 5 deaths each year. [3]. In cigarette smokers, the risk of developing lung cancer mainly depends on the number of years of the individual’s exposure. Other risk factors for lung cancer include secondhand smoke [4], HIV infection [5], and environmental risk factors [6]. Exposure to certain industrial substances such as arsenic, organic chemicals, radon, asbestos, radiation, air pollution, and environmental tobacco smoke are risk factors for the appearance of lung cancer in nonsmokers [7]. Some of these environmental risk factors are modifiable, and their removal can significantly lower the individual’s risk of developing lung cancer.

In smokers, after cessation of smoking, the risk of developing lung cancer still remains significantly higher than in nonsmokers for 15 years [8]. Accordingly, the prevention of lung cancer in ex-smokers represents a major unsolved problem for public health. Improvements in the early diagnosis of lung cancer have been made possible by imaging technologies such as spiral computer tomography. These methods are complex, expensive, and sensitive but poorly specific. A major problem limiting the secondary prevention of lung cancer (early diagnosis) is the lack of predictive intermediate biomarkers that may preliminarily identify high-risk subjects. Despite improved screening methods for lung cancer and advances in treatment, the 5-year survival rate is still only 20% [9]. 

MicroRNAs have been proposed as a possible predictive biomarker to deal with these problems. MicroRNAs are massively dysregulated during lung carcinogenesis induced by CS [10,11] and by other environmental airborne lung carcinogens [12]. MicroRNAs dysregulated in the lungs by CS and environmental pollutants are released extracellularly in the body fluids, including the blood, during the late stages of the carcinogenesis process, as demonstrated in mice for microadenomas, adenomas, and malignant tumors [13]. However, the translatability of these results to the human situation needs further study. In the blood, microRNA signatures predictive of lung cancer have been identified, but they vary greatly among different studies [14]. Many organs contribute to the microRNA burden in the blood, the contribution of the lungs being negligible as compared to those of the skeletal muscles or the liver. Only a low percentage of blood microRNAs originate from the lungs during smoke-induced carcinogenesis [15]. Accordingly, although encouraging results have been obtained, additional translation studies in preclinical models are still necessary in order to develop microRNAs as a predictive biomarker for lung cancer. 

With the development of next-generation sequencing, several targeted mutations and prognostic biomarkers for lung cancer patients have been discovered [16]. Different studies proved that let-7a is a better prognostic biomarker than classical biomarkers (E-cadherin, Snail, etc.) in NSCLC and other cancer types [17,18]. Tumor cell growth, invasion, and migration are slowed down by let-7a transfection into lung adenocarcinoma cells [18]. DNA methylation is one of the mechanisms that control let-7a expression. Let-7a-promoter hypomethylation increases the expression of let-7a, thus decreasing the development of lung adenocarcinomas cancerous cells [19]. Let-7a is downregulated in response to hypermethylation in epithelial cancers, and this downregulation is linked with poor prognosis [20]. Indeed, microRNAs play a pathogenic role in cancer only when the silenced oncogene is mutated, and the extracellular release of microRNAs corresponds to a cancer-related event and not to an adaptive response to carcinogen exposure. Exposure to environmental carcinogens, including CS, blocks microRNAs maturation by interfering with DICER [21]. MicroRNAs released from cancer cells are contained in exosomes and micro-vesicles that are used to communicate with other cells. This mechanism blocks specific immunity and activates epithelial–mesenchymal transition [22], inflammation, and tumor-associated macrophages to promote cancer growth [22]. MicroRNA release triggers inflammation because microRNA overload activates TLR3 receptors triggering cytokine production, protease release, leukocytes recruitment, and inflammation [23]. Release of extracellular vesicles plays a pathogenic role in the lung damage induced by CS [24]. 

Mutation of the KRAS oncogene is a pivotal pathogenic event in lung cancer. Recent studies show the activation of KRAS in colorectal (50%), pancreatic (90%), and lung adenocarcinomas (32%). In lung adenocarcinoma, KRAS has been observed to be mutated in codons 12, 13, and 61 [25]. The most frequent mutation observed was the substitution of glycine with cysteine and valine (G12C/G12V) in codon 12. These KRAS mutations are frequent in current and former smokers [25,26]. In smokers, these genomic alterations (KRAS mutations) are accompanied by epigenomic alterations mainly targeting the microRNA machinery. The expression of let-7a is downregulated in smokers as compared to nonsmokers [27]. This downregulation is associated with altered expression of inflammation mediators such as NF-κB in mammals [27]. In bronchial epithelial cells, exposure to CS downregulates let-7a expression [28] and increases cancer cell proliferation [29,30]. Exposure to polycyclic aromatic hydrocarbons contained in CS can induce glycine to tyrosine KRAS transversions in lung cancer [31]. The 3’UTRs of the RAS genes contain multiple let-7 complementary sites, allowing let-7 to regulate RAS expression [32,33]. Let-7 expression is lower, while RAS protein is higher, in lung tumors [33] as well as in radon induced lung damage [34]. Furthermore, a low expression of let-7 and a high expression of KRAS are correlated with the pathogenesis and prognosis of NSCLC [35]. Let-7 overexpression in lung cancer cells harboring KRAS mutation induces both cell cycle arrest and cell death, leading to suppression of tumor growth both in vitro and in vivo [32,36,37]. These findings provide direct evidence that let-7 acts as a tumor suppressor gene in lung cancer.

The aim of the present study is to evaluate the relationship between genomic altera-tion (oncogene KRAS mutations) and epigenomic alterations (let-7a expression) during CS-induced lung carcinogenesis. The parallel analysis of both KRAS and let-7a may represent a new lung cancer–predictive biomarker.

## 2. Results

### 2.1. Let-7a Expression 

CS downregulated let-7a expression in a time-dependent manner. In mice exposed to CS for 2 weeks, let- 7a expression was downregulated by 8.3% only as compared to the level of expression in the Sham-exposed mice (not significant), while in the mice exposed to CS for 8 months it was downregulated by 33.4% (*p* ≤ 0.01) (Figure 1). 

### 2.2. Let 7a Promoter Methylation 

DNA methylation of the let-7a promoter was analyzed 1 kb upstream of the transcription start site to evaluate its role, if any, in decreased let-7a expression. In the promoter, 2 CpG sites have been analyzed (red highlighted in Appendix A). As reported in the raw data file, the sequence analyses started at chr13:48,538,171-48,539,472. The first CpG site is located after 313 nucleotides and the second CpG site after further 16 nucleotides. Accordingly, the location of the first CpG site is at 48,538,484 and the location of the second CpG site is at 48,538,500. No significant difference was detected in the rate of let-7a promoter methylation between the Sham-exposed mice (55.1 ± 6.6 %) and the mice exposed to CS for either two weeks (53.1 ± 2.8 %) or eight months (51.5 ± 5.1%) (Figure 2).

### 2.3. KRAS Mutation 

The KRAS mutation amount significantly increased in the CS-exposed mice as compared to the Sham-exposed mice. The percentage of G/T transversions in codons 12 and 13 was 2.3 ± 0.40 % in the Sham-exposed mice, 6.4± 1.3 % in the mice exposed to CS for 2 weeks, and 11.5%± 2.9% in the mice exposed to CS for 8 months (Figure 3). These results indicate that CS exposure significantly increases KRAS mutation rate as compared to the Sham-exposed mice already after short-term exposure (2 weeks, *p* < 0.05), and even more after long-term exposure (8 months, *p* ≤ 0.01). Accordingly, KRAS mutation represents an early event in CS-induced carcinogenesis.

### 2.4. Cancer Incidence

The number of mice with tumors for each experimental group was with Sham, CS short, and CS long, and the number of mice is listed.

Cancer incidence was 6.5% in the Sham-exposed mice (4 cancer-bearing mice out of 61 animals), 3.4% in the mice exposed to CS for 2 weeks (CS short, 2 cancer bearing mice out of 60 animals), and 33.3% in the mice exposed to CS for 8 months (CS long, 21 cancer- bearing mice out of 63 animals). The cancer histotypes detected in the lungs of the mice exposed to CS for 8 months were benign adenoma (19%), unspecific adenocarcinoma (43%), papillary adenocarcinoma (33%), and acinar adenocarcinoma (5%) (Figure 4).

### 2.5. KRAS Mutations, Let-7a Expression, Let-7a Promoter Pethylation in Cancer-Bearing versus Cancer-Free Mice

No significant difference was detected comparing cancer-bearing (adenoma or adenocarcinoma) and cancer-free mice in either percentage of KRAS mutations (11.0 ± 2.1 vs. 11.6 ± 2.0), let-7a expression (0.030 ± 0.001 vs. 0.029 ± 0.001), or let-7a promoter methylation (52.5 ± 4.4 vs. 51.0 ± 6.3). 

Similarly, no difference in these biomarkers was detected comparing adenoma and adenocarcinoma.

The difference between the cancer-free and the cancer-bearing mice consisted in the relationship between the let-7a downregulation intensity and the KRAS mutation amount. Indeed, the relation between those two parameters was different between cancer-free and cancer-bearing when evaluated in the mice exposed to CS for 2 weeks. The relationship between these two variables was analyzed by linear regression analysis in the cancer-free mice as compared to the cancer-bearing mice, and in the mice exposed to CS for 8 months. The correlation between these two molecular alterations was significantly (*p* ≤ 0.01) positive in the cancer-free mice (Figure 5, upper panel) and significantly (*p* ≤ 0.05) negative in the cancer-bearing mice (Figure 5, lower panel).

## 3. Discussion

The obtained results indicate that CS induces time-dependent let-7a downregulation, as extensively reported in our previous papers. After 2 weeks of CS exposure, this downregulation is reversable in the case of smoking cessation, thus being referred to as ‘adaptive’ [38]. After 8 months of exposure, this downregulation is irreversible and represents cancer-prone molecular damage [39,40]. Our results demonstrate a barely detectable downregulation after 2 weeks of CS exposure, while a significant downregulation was detected in mice exposed to CS for 8 months. This finding suggests that let-7a downregulation as induced by CS requires long periods of exposure to become apparent. There the evaluation of the short-term effect of CS on let-7a expression underestimates this downregulation.

The sample size for the cancer incidence analysis (184, total mice, 64 mice exposed to CS for 8 months) was relatively small, which may limit the generalizability of the findings. However, the CS exposure model adopted, i.e., perinatal exposure, was highly effective in inducing lung cancer in CS-exposed mice [41], thus allowing for significative results using a relatively small number of mice.

The reported results shed further light on the mechanisms underlying microRNA downregulation as induced by CS. They indicate that CS does not induce alteration in let7a promoter gene methylation. This finding may be also due to the small sample size or to the relatively low sensitivity of the detection method used. Indeed, more sensitive methods such as chip-seq or intracellular staining by flow cytometry could be used to further explore this point. However, on the whole, the obtained results suggest that promoter methylation is less important than other mechanisms in inducing let-7a downregulation in CS-exposed mice.

The mechanisms causing let-7a downregulation consequent to CS exposure occur mainly at the post-genomic level. Our previous studies demonstrated that microRNA downregulation is caused by the binding of nucleophilic xenobiotic metabolites of CS to pre-miRNA, resulting in the formation of RNA adducts unable to be processed for further maturation by the DICER protein [12]. This event occurs far away from the nucleus in the cell cytoplasm, thus representing an epigenetic event. Another mechanism causing let-7a downregulation is the blockage of DICER catalytic pockets by the cumulative binding of CS electrophilic metabolites [38]. Let-7a downregulation induced by environmental pollutants and CS is a quite specific event. Indeed, age per se does not modify let-7a expression in the lungs [11].

None of the molecular alteration examined (oncogene KRAS mutation, let-7a gene methylation, let-7a expression downregulation) was able to distinguish between cancer- bearing and cancer-free mice even though each one of these events plays a pivotal role in carcinogenesis. This suggests that these biomarkers, when used alone, may not be useful for predicting lung cancer development. However, it cannot be excluded that other potential biomarkers may be more strongly associated with lung cancer development, notwithstanding the fact that these biomarkers are recognized as pivotal in lung carcinogenesis. Moreover, oncogenic KRAS expression has specifically been shown to induce aberrant DNA methylation, promoting hypomethylation across the genome while silencing key tumor suppressors through hypermethylation [39,41]. 

KRAS mutation is an early event consequent to CS exposure [38], and its frequency is low compared to microRNA downregulation. Our recent study of human lung cancer indicates that KRAS mutation occurred only in ≤10% of the patients while microRNA alteration was constantly present in all the cancer-bearing patients [42]. Oncogene mutation and microRNA expression are closely related events. The main task of microRNA is to suppress the expression of a mutated oncogene. This situation typically occurs for let-7a miRNA, recognizing messenger RNA produced by the mutated KRAS oncogene as one of its main targets [43]. Accordingly, the occurrence of KRAS mutation only is not sufficient to trigger the appearance of cancer, due to the efficacy of microRNA suppression. Let-7a downregulation per se does not represent a mechanism triggering cancer if no mutation in the oncogenes is present. This situation explains our results. Indeed, cancer-free mice may have a high level of KRAS mutation, but this is neutralized by a high level of let-7a expression. Conversely, the low expression of let-7a in these mice occurs mainly in the case of a low level of KRAS mutation. The lack of parallelism between a high level of KRAS mutation and a high intensity of let7 down regulation explains why cancer has not developed in these mice. Conversely, these molecular alterations are closely related in cancer-bearing mice. Indeed, our results provide evidence that cancer develops only in mice having a high level of KRAS oncogene mutation paralleled by a massive let-7a downregulation. Accordingly, the correlation between the rate of KRAS mutation and the intensity of the let7a downregulation represents a predictive hallmark of lung carcinogenesis in CS-exposed organisms. These results are in line with the fact that carcinogenesis is a complex process not related to single events but resulting from multiple molecular events targeting both the genome (KRAS) and the epigenome (let-7a), and explains why the same smoking treatment leads to different correlated results between let-7a and KRAS mutation in cancer-free and cancer-bearing patients.

A limitation of our study is that it was performed on mice, and the transferability of its results to human lung cancer needs to be demonstrated. However, microRNA is a highly conserved domain among different species [44]. Our previous studies specifically compared the alteration of microRNA induced by CS in humans and mice, demonstrating a good overlap of the results obtained [10].

Notwithstanding these limitations, our results provide new insights into the effects of CS exposure on lung tissue at the molecular level, which could have significant implications for the early detection and prevention of lung cancer.

In our opinion, the results may have interdisciplinary relevance going beyond cancer biology. Indeed, the parallel analysis of KRAS mutation and let-7a downregulation may represent a new molecular tool for identifying subjects at high risk for lung cancer to be selected for spiral TAC (early diagnosis and secondary prevention). Our findings are relevant for public health and CS control policy. Indeed, they shed new light on the molecular mechanisms linking CS to lung cancer, thus highlighting the importance of quitting smoking.

The parallel analysis of oncogene and microRNA could potentially pave the way for the development of new diagnostic tools or therapeutic strategies, thereby filling a critical gap in the current literature.

## 4. Materials and Methods

### 4.1. Animals

Mice lung samples were collected in our previous study [43]. A total of 184 strain H neonatal mice of Swiss albino mice (86 males and 98 females) were exposed whole-body to mainstream CS (MCS) for either 2 weeks or 8 months. Within 12 h after birth, the mice had been exposed to MCS under standardized conditions using a smoke exposure machine (model TE-10C, Teague Enterprises, Davis, CA, USA). The exposure was performed daily. Accordingly, 3 experimental groups were set up, including: (a) Sham-exposed mice, untreated mice kept in filtered air for 8 months (*n* = 61) (Sham); (b) mice exposed to MCS starting within 12 h after birth and continuing daily for 2 weeks (*n* = 60) (CS short); and (c) mice exposed to MCS, starting within 12 h after birth and continuing daily for 8 months (CS long) (*n* = 63) [43]. After 8 months, the mice were euthanized by slow CO_2_ asphyxiation, and the lungs were individually collected. A lung fragment (lower right lobe) was used for RNA extraction and microarray analysis. The lungs were fixed and subjected to standard histopathological analysis. The incidence and multiplicity of pre-neoplastic lesions (microadenomas) and lung tumors, distinguished according to their histopathological nature, were evaluated individually in each experimental group. Housing and treatments of the mice were carried out in accordance with European (2010/63 UE Directive) and institutional guidelines. The issuance of the Office of Laboratory Animal Welfare (OLAW) with the University of Genoa bears the identification number A5899-01.

As reported in Balansky et al. [40], the mice were exposed whole-body to MCS generated by commercially available cigarettes (Melnik King Size, Bulgartabac, Sophia, Bulgaria), having a declared content of 9.0 mg tar and 0.8 mg nicotine and delivering 10 mg CO each. The MCS was delivered to the exposure chambers by drawing 15 consecutive puffs, each puff being of 60 mL and lasting 6 s. Each daily session involved six consecutive exposures, lasting 10 min each, with 1 min intervals, during which a total air change was made. The average concentration of total particulate matter in the exposure chambers was 527 mg/m^3^.

### 4.2. DNA Extraction 

Genomic DNA was extracted from the lung tissue using the QIAamp DNA mini-Kit (Qiagen, Hilden, Germany) according to the manufacturer's instructions. The DNA was extracted from the tissue specimens (50 mg) collected individually from the lungs of 184 mice. The specimens were homogenized by using a tissue lyser (Tissue Lyser, Qiagen, Gaithersburg, MD, USA) and, after centrifugation at 14,000× *g* at 4 °C for 15 min, the DNA was purified from the supernatant by using a commercial kit (GenElutt™ Mammalian Genomic DNA Miniprep kit, Sigma, St. Louis, MO, USA). Spectrophotometric analyses by a fiber optic spectrophotometer (Nanodrop ND-1000, Thermo Scientific, Wilmington, DE, USA) showed the quality of the DNA extracted from the 184 samples, with an average 260/280 ratio of 1.9 ± 0.01 (mean ± SE). Subsequently, 500 ng of each DNA sample was treated with bisulphite using an EZ DNA Methylation-Gold Kit (D5007; Zymo Research, Orange, CA, USA). The bisulphite-treated DNA was eluted in 30 μL of M-Elution Buffer (D5001-6, Zymo Research) and diluted in further 170 μL of purified water.

### 4.3. RNA Extraction 

The lung fragments (10 mg each) were homogenized in QIAzol Lysis Reagent (700 μl) by continuous shaking in the tissue lyser for 2 min at 30 Hz. The homogenates were centrifuged at 14,000× *g* at 4 °C for 15 min to remove cell debris, and the pulmonary miRNAs were purified from the supernatants by using an miRNeasy kit (Qiagen, Hilden, Germany). The miRNA amounts were standardized among the blood serum samples for RT- PCR analyses using Qubit™ 3.0 Fluorometer (Life Technologies, Gent, Belgium).

### 4.4. Bisulphite Treatment and DNA Methylation Analyses

DNA methylation was quantified using bisulphite-PCR and pyrosequencing. A 50 μL PCR was carried out using 25 μL GoTaq Green Master Mix (Promega, Madison, WI, USA), 20 pmol each of the forward and reverse primers, 25 ng bisulphite-treated genomic DNA, and water [44]. The assays were designed to cover the greatest possible number of CpG sites within the let-7a-promoter region considering PCR amplicon length, target sequences, and primers that avoided CpG-enriched regions. 

### 4.5. Computational Analysis of the Let-7a Promoter

The DNA sequence of the let-7a was obtained from the National Center for Biotechnology Information (93 nucleotides from 48538179 to 48538272 on chromosome 13, Accession NR_029725.1). The promoter sequence (1-kb upstream of the let-7a sequence) was determined using the BLAT genome browser gateway (National Center for Biotechnology Information Assembly MGSCv37). The DNA sequence was then analyzed with PROMO software (Version 8.3) to identify putative binding sites for different transcription factors. E2F2 and NFκB binding sites upstream of the let-7a start site were identified.

### 4.6. Let-7a Expression

The total RNA (10 ng) was reverse-transcribed using miR-specific stem-loop RT primers (TaqMan MicroRNA Assays; Applied Biosystems, Thermo-Fisher, Waltam, MA, USA) and components of a high-capacity cDNA reverse transcription kit (Life Technologies, Carlsbad, CA, USA) according to the manufacturer’s protocols. The expression levels of the let-7a were detected by RT-PCR using TaqMan MicroRNA assays (Life Technologies, Carlsbad, CA, USA) and a QuantStudio™ 1 Real-Time PCR System (Applied Biosystems, Thermo-Fisher, Waltam, MA, USA) with standard thermal cycling conditions, in accordance with manufacturer recommendations. PCR reactions were performed in triplicate in the final volumes of 20 µL, including inter-assay controls (IAC) to account for variations between runs. RT-PCR (TaqMan MicroRNA Assays; Applied Biosystems, Thermo-Fisher) was used to quantify the expression of the let-7a according to the manufacturer’s instructions. To normalize the data for quantifying miRNAs, the universal small nuclear RNU38B (RNU38B Assay ID 001004; Applied Biosystems, Thermo-Fisher, Waltam, MA, USA) as an endogenous control was used [45]. The delta–delta Ct method was used to calculate the variation in let-7a expression between the various treatments. The CS short and CS long groups were compared with the control group (Sham). The quantitative results of the let-7a expression were inferred from a comparison of the PCR results with the calibration curve set up using the RNU38B reference standard and expressed as copies per microliter. 

Each 15 μL of the reaction system contained 0.15 μL of 100 mM dNTPs with dTTP, 1 μL of MultiScribe Reverse Transcriptase (50 U/μL), 1.5 μL of RT buffer (×10), 0.1 μL of RNase inhibitor (20 U/μL), 6.25 μL of nuclease-free water, 5 μL of small RNA, and 3 μL of RT primer. The microRNA was quantified by a Qubit 3 fluorimeter (Life Technologies, Carlsbad, CA, USA). The thermal cycling conditions were 30 min at 16 °C, 30 min at 42 °C, and 5 min at 85 °C. Each 20 μL of the reaction system for real-time quantitative PCR contained 1 μL of real-time primer, 1.33 μL of the product from the RT reaction, 10 μL of TaqMan Universal PCR Master Mix, and 7.67 μL of nuclease-free water. The reactions were performed in triplicate on a QuantStudio™ 1 Real-Time PCR System (Applied Biosystems, Thermo-Fisher, Waltam, MA, USA) for 10 min at 95 °C, followed by 45 cycles of 15 s at 95 °C and 1 min at 60 °C. Along with the Cq values calculated automatically by the SDS software (V1.2.x) (threshold value = 0.2, baseline setting: cycles 3–15), raw fluorescence data (Rn values) were exported for further analyses.

### 4.7. KRAS Mutation by Pyrosequencing

A polymerase chain reaction (PCR) of exon 2 to detect KRAS codon 12 and 13 mutations was performed using the following primers: 5′-CGATGGAGGAGTTTGTAAATGAA-3’ and 5′-/BioTEG/TTCGTCCACAAAATGATTCTGA-3′. A PCR amplification was performed for 55 cycles with an annealing temperature of 58 °C. The PCR products were analyzed using pyrosequencing with the Pyromark MD (Qiagen) using the internal primer 5′-AAACTTGTGGTAGTTGGA-3′. Every sample was tested three times for each marker to confirm reproducibility and increase the precision of the results. The average of the three replicates was used in statistical analyses.

KRAS G/C transversion at codon 12 and 13 was selected because this is the most common mutation in lung cancer as inferred from the available literature and the COSMIC database (https://cancer.sanger.ac.uk/cosmic, accessed on 1 January 2020).

### 4.8. Statistics

Comparisons between the groups regarding the survival of the mice and the incidence of histopathological lesions were made by Chi-square analysis. Comparisons between the groups regarding exposure to CS were made by ANOVA and the Student’s t test for unpaired data. Since the tumor multiplicity data did not have a normal distribution, the effects of gender and the exposures were assayed using the Kruskal–Wallis rank-sum test, and pairwise comparisons between groups were carried out by the Wilcoxon rank-sum test. The estimated *p* values were adjusted for multiple comparisons by using the Bonferroni post hoc test. All the statistical analyses were performed using the statistical software Statview (Version 4.0) (Abacus Concept Inc., Berkeley, CA, USA). The qPCR data were expressed as means ± SD of 3 replicates, and differences between the groups were evaluated by the Student’s *t* test for unpaired data.

## 5. Conclusions

Our findings provide experimental evidence demonstrating a concomitant role of KRAS oncogene mutation and massive let-7a downregulation in lung carcinogenesis induced by CS exposure. The contemporary analysis of both genetic and epigenetic damages may represent a relevant and reliable approach in preventive and predictive medicine to identify early subjects at high risk for lung cancer development. In cigarette smoke–exposed mice, the correlation between let-7a expression level and the number of KRAS transversion at codons 12 and 13 was opposite between the cancer-free and cancer-bearing mice. This result provides evidence that both genetic and epigenetic alterations are required to produce lung cancer in CS-exposed organisms. Various miRNAs appear to be promising therapeutic targets from a scientific perspective [46]. 

The delivery system should be optimized to deliver the exogenous miRNA and for it to exert its effect by integrating with the genome. miRNA-based therapy could also be used in conjunction with already established treatments such as chemotherapy or radiation. The ability of certain miRNAs in chemo sensitization would be beneficial in combination therapies. Several in vitro and in vivo studies have shown the effectiveness of miRNA therapeutics in reversing certain hallmarks of cancer. Several preclinical and clinical studies are ongoing, and in the future, we could expect to develop miRNA sequences based on an individual genome to provide personalized cancer therapy. 

## Figures and Tables

**Figure 1 ijms-24-11778-f001:**
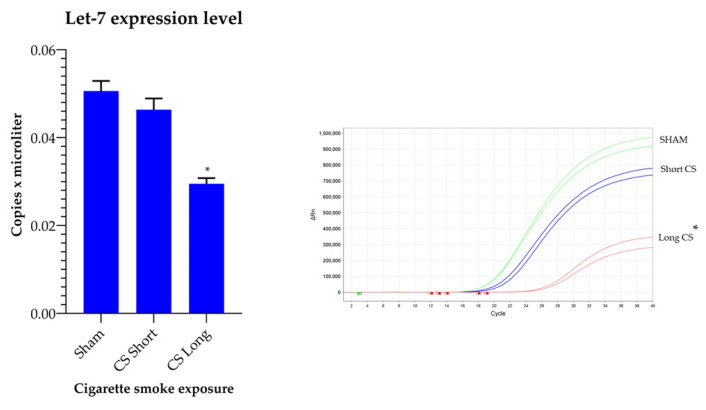
Quantification of let-7a expression intensity calculated by referring to RNU38B reference standard (vertical axis, copies per microliter) as valuated by qPCR in mice either unexposed to CS (Sham) or exposed for 2 weeks (CS short) and for 8 months (CS long). * *p* < 0.01 as evaluated by t-Student test by unpaired data CS long exposure vs. Sham and CS long vs. CS short exposure.

**Figure 2 ijms-24-11778-f002:**
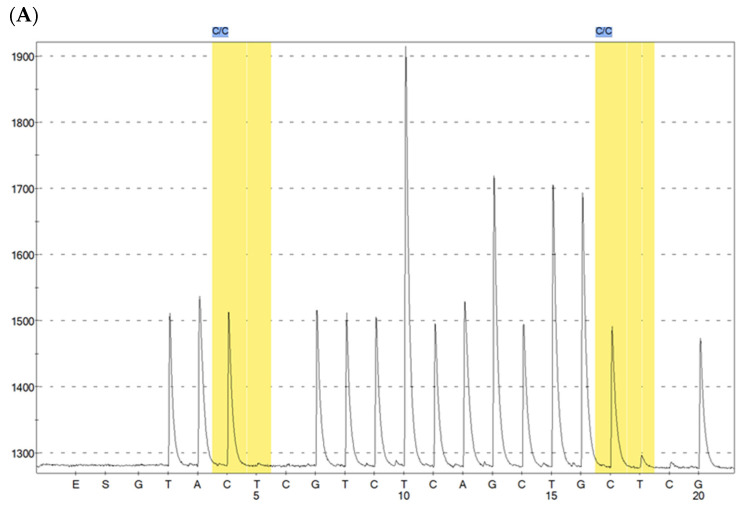
(**A**) example of pyrogram trace illustrating the analysis of mouse let-7 promoter. (**B**) example of pyrogram trace illustrating the percentages of CpG methylation for the two analyzed CpG positions of mouse let-7 promoter (**C**) Methylation percentage of let-7a promoter gene (vertical axis) in mice either unexposed to CS (Sham) or exposed for 2 weeks (CS short) and for 8 months (CS long).

**Figure 3 ijms-24-11778-f003:**
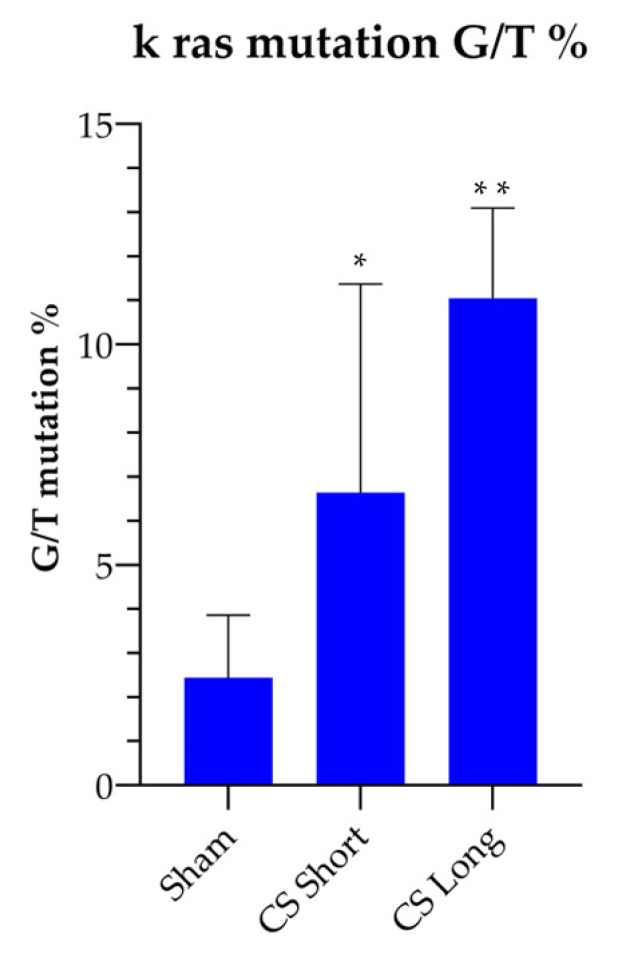
Pyrosequencing analysis of KRAS gene mutation (percentage of G/T transversions in codons 12 and 13) in DNA isolated from lungs of mice either unexposed (Sham) or exposed to CS for 2 weeks (short CS) or 8 months (long CS). * *p* < 0.05 vs. Sham, ** *p* ≤ 0.01 vs. Sham.

**Figure 4 ijms-24-11778-f004:**
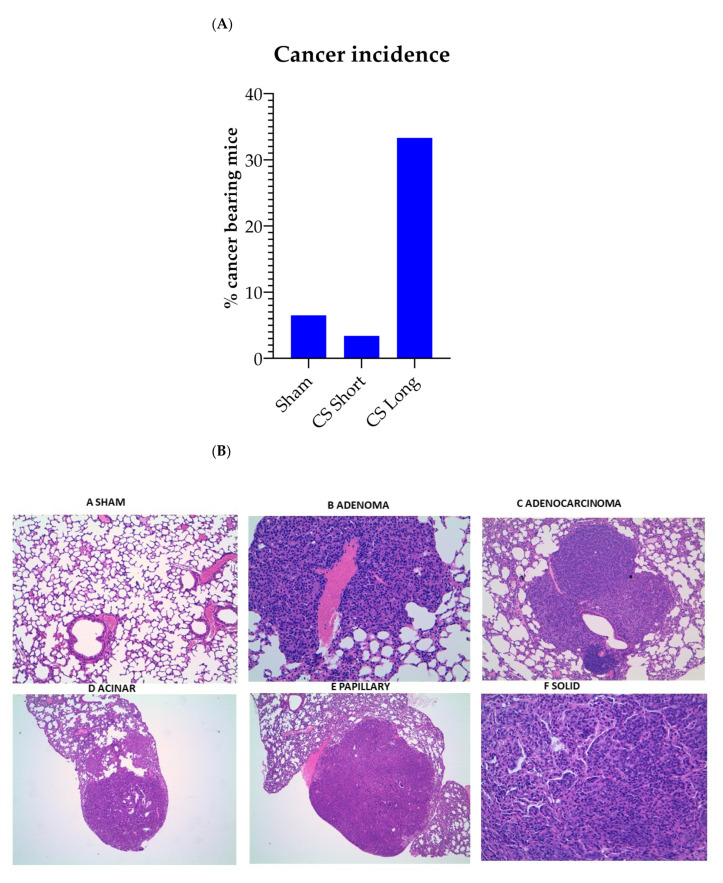
(**A**) cancer incidence in mice belonging to the 3 experimental groups. (**B**) upper panel, microscope morphology (A: Sham magnification 25×; B: adenoma 100×; C: adenocarcinoma 25×; D: acinar 25×; E: papillary 25×; F: solid 200×); (**C**) number of cancers distributed according to their histotypes as detected in the lungs of mice exposed to cigarette smoke for 8 months.

**Figure 5 ijms-24-11778-f005:**
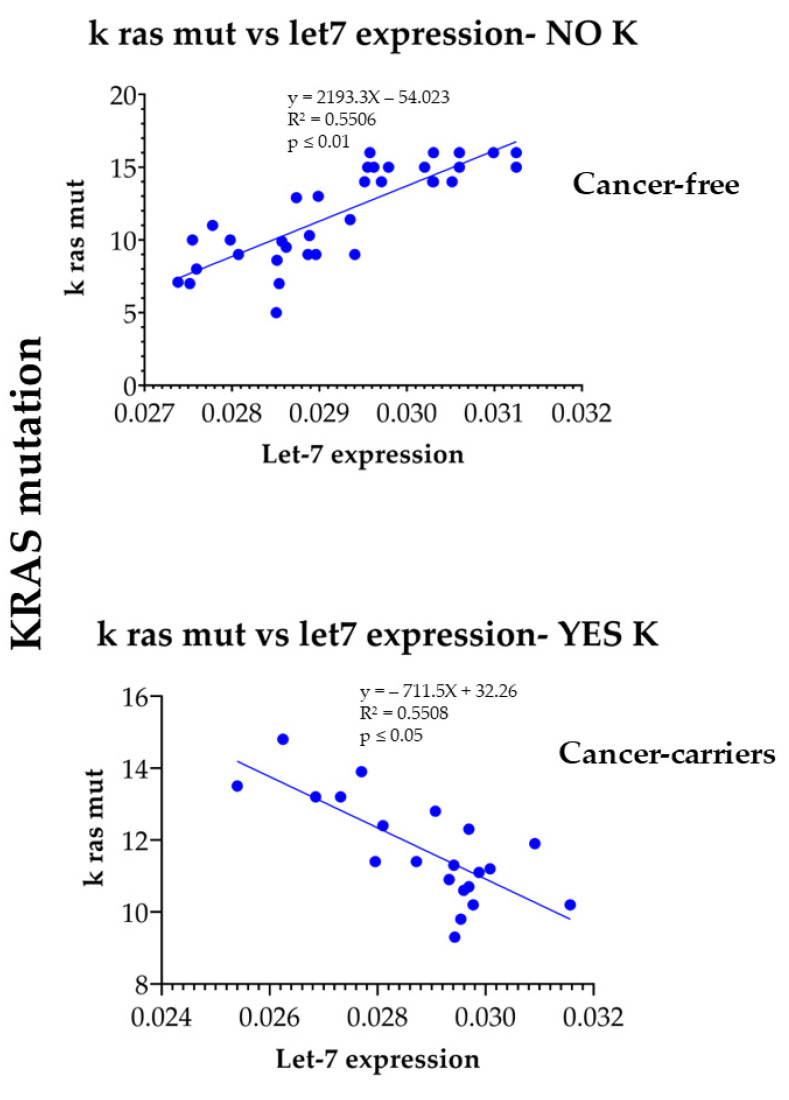
Regression analysis of the relationship between KRAS mutation amount (vertical axis) and let-7a expression (horizontal axis) in cancer-free (**upper** panel) and cancer-bearing mice (**lower** panel), both exposed to CS for 8 months.

## Data Availability

The datasets used and/or analyzed during the present study are available from the corresponding author on reasonable request.

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
