# Peer review of "Let-7a Downregulation Accompanied by KRAS Mutation Is Predictive of Lung Cancer Onset in Cigarette Smoke–Exposed Mice"

_ijms, 2023, doi:10.3390/ijms241411778_

Round 1

Reviewer 1 Report (Previous Reviewer 1)

Upon reviewing the study, it became apparent that several key aspects were not adequately addressed or clarified, particularly in relation to my first comments.

Minor editing of English language required

Author Response

Dear Editors,

We would like to thank you for considering the manuscript entitled “Let-7a downregulation accompanied by KRAS mutation is predictive of lung cancer onset in Cigarette Smoke exposed mice” by Pulliero A. et al., and for sharing the Reviewers’ comments that certainly helped in improving the quality of the manuscript (ID: ijerph-2445830). We appreciated the Reviewers’ comments, and we revised the manuscript accordingly. Please find enclosed to the submission of the revised version of the manuscript the point-by point reply to the Reviewers’ comments.

We hope that this revised version of our MS will be now suitable for publication in the IJERPH.

Accordingly, we prepared a revised version of the manuscript acknowledging Referees’ and Editor’s comments as below specified:

Editor Comments: Figures are still badly drawn with Excel. Asterisks should be added above the columns to indicate statistical significance when there is.

Answer: We thank the Editor for the note. Figures have been revised as requested and the Asterisks have been added.

Reviewer 1

COMMENT 1. Minor editing of English language required

ANSWER 1. We has been revised the English language as requested.  

Reviewer 2 Report (New Reviewer)

Major comments:

1.     There is no methylation on the Let7a promoter. What is the mechanism causing Let-7a downregulation by smoking.

2.     There is no methylation on the Let-7a promoter. Please delete the DNA methylation part in the graphic abstract.

3.     Please show statistical analysis in Figure 3.

4.     Let-7a can be further detected in the tumor tissues compared to SHAM group by in situ hybridization to confirm the findings.

5.     There is no difference of KRAS mutation and Let7a expression between cancer-free and cancer-bearing mice in Figure 5, which may indicate that KRAS mutation and Let7a expression are not the major genes causing cancer in this model.

6.     Please show statistical analysis in Figure 6.

7.     Why the same smoking treatment leads to totally different correlated results between Let-7a and KRAS mutation in Cancer-free and Cancer-carriers, respectively, in Figure 6.

Minor comments:

1.     What is the unit of Y-axis in Figure 1.

2.     Please show the unit of Y-axis in Figure 4A.

Author Response

Reviewer 2

COMMENT 1. There is no methylation on the Let7a promoter. What is the mechanism causing Let-7a downregulation by smoking.

ANSWER 1. A paragraph describing the mechanisms causing let-7a downregulation by smoking has been added in Disucssion (lines 224-231).

COMMENT  2. There is no methylation on the Let-7a promoter. Please delete the DNA methylation part in the graphic abstract.

ANSWER 2. The DNA methylation part has been delete in the GA as requested.

COMMENT  3.    Please show statistical analysis in Figure 3.

ANSWER 3. The statistical analysis has been added. 

COMMENT 4.   Let-7a can be further detected in the tumor tissues compared to SHAM group by in situ hybridization to confirm the findings.

ANSWER 4. We thank the Reviewer for the note. Unfortunately, the biological material available is not sufficient to be able to carry out this investigation.

COMMENT 5. There is no difference of KRAS mutation and Let7a expression between cancer-free and cancer-bearing mice in Figure 5, which may indicate that KRAS mutation and Let7a expression are not the major genes causing cancer in this model.

ANSWER 5. We thank the Reviewer for the note. We have delete Figure 5 as suggested before.

COMMENT 6.   Please show statistical analysis in Figure 6.

ANSWER 6. We thank the Reviewer for the note. We have been added the statistaical analysis as suggested.  

COMMENT 7.   Why the same smoking treatment leads to totally different correlated results between Let-7a and KRAS mutation in Cancer-free and Cancer-carriers, respectively, in Figure 6.

ANSWER 7. The ansewer to this question is detailed in Dicussion (lines 272-296). A center further clarifying this point has now been added in Discussion (lines 294-296)

COMMENT 8.   What is the unit of Y-axis in Figure 1.

ANSWER 8. The Unit of Y-axis has been added. 

COMMENT 9. Please show the unit of Y-axis in Figure 4A.

ANSWER 9. The Y-axis in Figure 4A has been added

Reviewer 3 Report (New Reviewer)

General comments to the paper entitled: Let-7a downregulation accompanied by KRAS mutation is predictive of lung cancer onset in Cigarette Smoke exposed mice

The study aimed to investigate the possible correlation between the KRAS mutation in mice exposed to cigarette smoke and the let-7a microRNA expression. The Let-7a expression significantly decreased after 8 months of exposure to cigarette smoke but the methylation did not show any changes. It was confirmed that the KRAS mutation significantly increased in cigarette smoke-exposed mice. According to the study, the incidence of cancer increases if the two events are present at the same time: a high level of KRAS mutation and the downregulation of Let-7a expression. 

line 131: delete “a” from Fig 1a

line 136: The first sentence is duplicated. Please delete it. Unified the indication of let-7a. Indicate “*” in Figure 1.

line 145: Replace 9 months with 8 months.

line 170: Fig.3 shows the changes in % of the mutation, not the number.

line 172: In Fig 4 indicate “%”.

line 194: As no differences in KRAS mutation, let-7a expression, and let-7a promoter methylation, I suggest deleting Fig 5.

line 226: I agree that the sample size was relatively small.

line 429: Edit the References 1-11.

Author Response

Reviewer 3.

COMMENT 1. line 131: delete “a” from Fig 1a

line 136: The first sentence is duplicated. Please delete it. Unified the indication of let-7a. Indicate “*” in Figure 1.

line 145: Replace 9 months with 8 months.

line 170: Fig.3 shows the changes in % of the mutation, not the number.

line 172: In Fig 4 indicate “%”.

line 194: As no differences in KRAS mutation, let-7a expression, and let-7a promoter methylation, I suggest deleting Fig 5.

line 226: I agree that the sample size was relatively small.

line 429: Edit the References 1-11.

ANSWER 1. We thank the Reviewer for the revisions. All the revisions requesed have been reported in the text marked in yellow.

Reviewer 4 Report (New Reviewer)

Following are my comments for the manuscript:

1) The abstract has serious errors with respect to case errors, grammar errors as well as massive inconsistency in addressing Let-7a. (Certain places let7, let7a or let-7a). Please be precise and consistent. Grammar and sentence formation errors throughout lines 21-25 and last lines of abstract. Need to re-write; consider breaking the sentences. 

2) There are grammar and language errors throughout introduction for example, line 49-51, 53-56, 80-82, 106-109, 103 to name a few. It needs major corrections and language re-writing for example in line 117 "cancer proliferation".. I have never heard this term being used; Sentence formation and language errors in lines 121-125

3) Figure 1  legends' opening sentence is repetitive; Unit of Let-7a expression is missing on Y-axis; please indicate P-values on graph; Let-7a expression for control animals at early stage and late stage is required to draw fair comparison

4) For Figure 2, non-significance should be indicated on graphs, controls for both time points are required; did methylation pattern change in control mice also with age?

5) Language errors in description of figure 3 in line 164-165; typos in line 171-172 (consider re-writing)

6) Please comment on more cancer incidence in sham control mice than short CS exposed mice in figure 4A; label of Y-axis is missing in 4A; Please quantify 4B; in 4C, figure axis shows mice numbers and figure legends shows mice %

7) Language errors in figure 5 description in line 191; label of Y-axis is missing in figure 5

8) In line 202, "item" is very inappropriate word; typo in line 213

9) Line 216-230 is full with language and grammar errors; re-writing is mandatory; Have authors considered analysis of methylation by high sensitivity methods such as chip-seq or intracellular staining by flow cytometry (with reference to lines 238-240)?; heavy grammar errors in line 249, 254, 266, 273 and so forth; Line 278-288 is well written

10) In method section, line 291-294 has language errors, line 302 needs to be re-written; please re-write lines 331-337 as it is confusing; typo in line 352 as it reads RQ-PCR instead of RT-PCR; error in line 358; language error in line 395

11) Conclusion is decent

12) I recommend to prepare figures using Graphpad Prism for better quality and in interest of journal

Please refer to my comments in above section. This manuscript is poorly written and it needs heavy editing

Author Response

Reviewer 4

COMMENT 1. The abstract has serious errors with respect to case errors, grammar errors as well as massive inconsistency in addressing Let-7a. (Certain places let7, let7a or let-7a). Please be precise and consistent. Grammar and sentence formation errors throughout lines 21-25 and last lines of abstract. Need to re-write; consider breaking the sentences.

ANSWER 1. We thank the Reviewer for the comments. We have fully rewritten the abstract as suggested.

COMMENT 2.  There are grammar and language errors throughout introduction for example, line 49-51, 53-56, 80-82, 106-109, 103 to name a few. It needs major corrections and language re-writing for example in line 117 "cancer proliferation". I have never heard this term being used; Sentence formation and language errors in lines 121-125.

ANSWER 2. We thank the Reviewer for the note. The Introduction was fully rewritten.

COMMENT 3. Figure 1  legends' opening sentence is repetitive; Unit of Let-7a expression is missing on Y-axis; please indicate P-values on graph; Let-7a expression for control animals at early stage and late stage is required to draw fair comparison

ANSWER 3. We thank the Reviewer for the note. We extensively analyzed the influence of both age and CS on let-7a in an our previously published paper demonstrating that only CS but not age affects let-7a expression (Izzotti et al., Relationship of microRNA expression in mouse lung with age and exposure to cigarette smoke and light. FASEB J., 23: 3243-3250 (2009). A sentence explaining this isuue has been added in Disucssion (lines 245-247).

COMMENT 4. For Figure 2, non-significance should be indicated on graphs, controls for both time points are required; did methylation pattern change in control mice also with age?

ANSWER 4.  We thank the Reviewer for the comments. See reponse to comment 3.

COMMENT 5. Language errors in description of figure 3 in line 164-165; typos in line 171-172 (consider re-writing).

ANSWER 5. We thank the Reviewer for the note. We have been revised the sentences, as requested.

COMMENT 6.  Please comment on more cancer incidence in sham control mice than short CS exposed mice in figure 4A; label of Y-axis is missing in 4A; Please quantify 4B; in 4C, figure axis shows mice numbers and figure legends shows mice %

ANSWER 6. We thank the Reviewer for the note. We have added the label to the figures as requested.

COMMENT 7. Language errors in figure 5 description in line 191; label of Y-axis is missing in figure 5

ANSWER 7. We thank the Reviewer for the note. The Figure 5 has been delete as requested before.

COMMENT 8. In line 202, "item" is very inappropriate word; typo in line 213

ANSWER 8. We thank the Reviewer for the note. We have been corrected the sentence.

COMMENT 9. Line 216-230 is full with language and grammar errors; re-writing is mandatory;

Have authors considered analysis of methylation by high sensitivity methods such as chip-seq or intracellular staining by flow cytometry (with reference to lines 238-240)?; heavy grammar errors in line 249, 254, 266, 273 and so forth; Line 278-288 is well written.

ANSWER 9. We thank the Reviewer for the note. These paragrahs have been fully re-written and  reviwers comment added (Discussion, lines 224-242).

COMMENT 10. In method section, line 291-294 has language errors, line 302 needs to be re-written; please re-write lines 331-337 as it is confusing; typo in line 352 as it reads RQ-PCR instead of RT-PCR; error in line 358; language error in line 395

ANSWER 10. We thank the Reviewer for the note. We have corrected the method section as requested.

COMMENT 11.  Conclusion is decent

ANSWER 11. We thank the Reviewer for the comment.

COMMENT 12. I recommend to prepare figures using Graphpad Prism for better quality and in interest of journal

ANSWER 12. We thank the Reviewer for the note.

Reviewer 5 Report (New Reviewer)

I would like to thank the handling editor for giving me the opportunity to review the manuscript entitled “Let-7a downregulation accompanied by KRAS mutation is predictive of lung cancer onset in cigarette smoke exposed mice” by Pulliero and colleagues, which is currently under consideration for publication in the International Journal of Molecular Sciences. I would also like to commend the authors for their scholarly work, which presents a comprehensive investigation into the molecular mechanisms of lung cancer, with a particular focus on the role of microRNAs and the impact of cigarette smoke exposure.  The study utilized a cohort of 184 Swiss albino mice, which were exposed to mainstream cigarette smoke (MCS) for varying durations. The experimental groups included mice exposed to MCS for two weeks, mice exposed for four months followed by four months in filtered air, and a control group kept in filtered air for eight months. The exposure to MCS was carefully controlled, with the average concentration of total particulate matter in the exposure chambers being 527 mg/m3. The researchers extracted genomic DNA and microRNAs from lung tissue samples, using advanced techniques, such as bisulphite treatment and pyrosequencing, to analyse DNA methylation. The study also examined the expression of specific microRNAs, including let-7a, within the lung tissue. The findings of the study contribute to the understanding of the molecular changes induced by cigarette smoke, including alterations in microRNA expression and DNA methylation patterns. The research also underscores the potential of microRNAs as biomarkers for pulmonary tumorigenesis in cigarette smoke-exposed mice. Finally, the authors reference several previous studies, highlighting the role of microRNAs in the context of age and exposure to cigarette smoke, the effects of environmental chemical carcinogens on microRNA machinery, and the modulation of genomic and epigenetic endpoints by certain drugs.

The manuscript situates itself within the existing body of literature on lung cancer, particularly in the field of molecular oncology. It has the potential to contribute to the ongoing discourse on the role of microRNAs in carcinogenesis, a topic that has garnered significant attention in recent years due to the potential of microRNAs as diagnostic and therapeutic targets. The study's focus on the impact of cigarette smoke exposure on microRNA expression and DNA methylation patterns in lung tissue is particularly noteworthy. While the deleterious effects of cigarette smoke on lung health are well-documented, the present experimental study delves into the molecular mechanisms underlying these effects, thereby enriching our understanding of how cigarette smoke contributes to lung cancer development. The use of a comprehensive mouse model, coupled with careful experimental design and robust analytical techniques, lends credibility to the study. The findings provide valuable insights into the early and late effects of cigarette smoke exposure on lung tissue at the molecular level, which could have significant implications for the early detection and prevention of lung cancer. Moreover, the study's exploration of the potential of microRNAs as biomarkers for pulmonary tumorigenesis in cigarette smoke-exposed mice introduces a novel avenue for future research. This could potentially pave the way for the development of new diagnostic tools or therapeutic strategies, thereby filling a critical gap in the current literature.

Nevertheless, there are a few areas where the manuscript could be improved to enhance its quality and impact:

1.      While the manuscript provides a detailed description of the experimental design, it would be beneficial to further clarify certain aspects of the methodology. For instance, the criteria for selecting the specific microRNAs for analysis could be elaborated upon. Additionally, the rationale behind the chosen durations of cigarette smoke exposure could be explained in more detail.

2.      The manuscript could benefit from a more explicit discussion of the study's limitations. For example, the authors could acknowledge the limitations of using a mouse model and discuss the potential challenges in translating these findings to human lung cancer.

3.      The discussion section could provide a broader contextualization of the findings within the current literature. This could include a comparison of the study's findings with those of previous studies, as well as a discussion of how the findings contribute to the current understanding of lung cancer pathogenesis.

4.      The authors could elaborate more on the potential implications of their findings for the diagnosis and treatment of lung cancer. They could also suggest specific future research directions based on their findings, such as further investigation into the role of specific microRNAs as potential therapeutic targets.

5.      The authors could consider briefly discussing the interdisciplinary relevance of the study. For instance, the implications of the findings for public health policy or tobacco control efforts could be explored. This could broaden the manuscript's appeal and increase its potential impact.

6.      The conclusion section could be strengthened by summarizing the key findings more succinctly and clearly stating their significance. This would provide a strong ending to the manuscript and leave a lasting impression on the reader.

In conclusion, I would like to reiterate my appreciation to both the editor and the authors for the opportunity to review this interesting and informative manuscript. I believe that the suggested modifications, if addressed, will further enhance the quality and impact of the work. I wish the authors success in their ongoing research endeavours.

The authors demonstrate a proficient use of academic language, with clear and concise sentences that effectively communicate complex scientific concepts. The vocabulary is appropriate for the subject matter and the intended audience, which is primarily academic and professional. However, there are a few areas where the language could be improved:

1.      Some sentences could be restructured for better clarity and readability. For instance, in the sentence “For the specific microRNA let7a the assays were designed to cover the greatest possible number of CpG sites within the promoter region but considering length of the PCR amplicon, length of the target sequence and primers that avoided CpGs”, the phrase “but considering” is somewhat ambiguous in this context, and it is not entirely clear how it relates to the rest of the sentence.

2.      The authors may consider ensuring consistency in the use of terminology throughout the manuscript. For example, since “cigarette smoke” is abbreviated as “CS”, this abbreviation should be used consistently throughout the manuscript.

3.      While the grammar appears to be generally good, there are minor errors that could be corrected. For instance, the authors may wish to ensure correct use of articles (a, an, the) and prepositions (of, in, on). Also, they could check for correct punctuation, particularly in complex sentences.

Author Response

Reviewer 5.

COMMENT. The manuscript situates itself within the existing body of literature on lung cancer, particularly in the field of molecular oncology. It has the potential to contribute to the ongoing discourse on the role of microRNAs in carcinogenesis, a topic that has garnered significant attention in recent years due to the potential of microRNAs as diagnostic and therapeutic targets. The study's focus on the impact of cigarette smoke exposure on microRNA expression and DNA methylation patterns in lung tissue is particularly noteworthy. While the deleterious effects of cigarette smoke on lung health are well-documented, the present experimental study delves into the molecular mechanisms underlying these effects, thereby enriching our understanding of how cigarette smoke contributes to lung cancer development. The use of a comprehensive mouse model, coupled with careful experimental design and robust analytical techniques, lends credibility to the study. The findings provide valuable insights into the early and late effects of cigarette smoke exposure on lung tissue at the molecular level, which could have significant implications for the early detection and prevention of lung cancer. Moreover, the study's exploration of the potential of microRNAs as biomarkers for pulmonary tumorigenesis in cigarette smoke-exposed mice introduces a novel avenue for future research. This could potentially pave the way for the development of new diagnostic tools or therapeutic strategies, thereby filling a critical gap in the current literature.

ANSWER. We wish to thank the Reviewer for the kind and positive comments. We have revised the manuscript, taking into account the Reviewer’s concerns below.

COMMENT 1.    While the manuscript provides a detailed description of the experimental design, it would be beneficial to further clarify certain aspects of the methodology. For instance, the criteria for selecting the specific microRNAs for analysis could be elaborated upon. Additionally, the rationale behind the chosen durations of cigarette smoke exposure could be explained in more detail.

ANSWER 1. We are grateful with the Reviewer for the suggestions. A sentence has been added in Intriduction to better explain the criteria for selecting monitored end points (Introduction, last paragraph, lines 116-120).

COMMENT 2. The manuscript could benefit from a more explicit discussion of the study's limitations. For example, the authors could acknowledge the limitations of using a mouse model and discuss the potential challenges in translating these findings to human lung cancer.

ANSWER 2. We thank the Reviewer for the note. A paragraph (including 1 newly added referencea) reporting the limits of using a mouse model and the problem of results translaboility to human lung cancer has been added in Discucssion (lines 284-290).

COMMENT 3. The discussion section could provide a broader contextualization of the findings within the current literature. This could include a comparison of the study's findings with those of previous studies, as well as a discussion of how the findings contribute to the current understanding of lung cancer pathogenesis.

ANSWER 3. We thank the Reviewer for the suggestion. A paragraph has been added in the discussion as requested. 

COMMENT 4. The authors could elaborate more on the potential implications of their findings for the diagnosis and treatment of lung cancer. They could also suggest specific future research directions based on their findings, such as further investigation into the role of specific microRNAs as potential therapeutic targets.

ANSWER 4. We thank the Reviewer for the note. We have added the paragraph in the Conclusion section.

COMMENT 5.   The authors could consider briefly discussing the interdisciplinary relevance of the study. For instance, the implications of the findings for public health policy or tobacco control efforts could be explored. This could broaden the manuscript's appeal and increase its potential impact.

ANSWER 5. We thank the Reviewer for the note. A sentence highlihting the interdisciplinary relevance has been added in Discussion (lines 294-299)

COMMENT 6. The conclusion section could be strengthened by summarizing the key findings more succinctly and clearly stating their significance. This would provide a strong ending to the manuscript and leave a lasting impression on the reader.

ANSWER 6. The conclusion section has been improved.

COMMENT 7. In conclusion, I would like to reiterate my appreciation to both the editor and the authors for the opportunity to review this interesting and informative manuscript. I believe that the suggested modifications, if addressed, will further enhance the quality and impact of the work. I wish the authors success in their ongoing research endeavours.

ANSWER 7. We thank the Reviewer for the suggestion. We have added a paragraph in the conclusion as suggested.

COMMENT 8. Comments on the Quality of English Language

The authors demonstrate a proficient use of academic language, with clear and concise sentences that effectively communicate complex scientific concepts. The vocabulary is appropriate for the subject matter and the intended audience, which is primarily academic and professional. However, there are a few areas where the language could be improved:

ANSWER 8. We wish to thank the Reviewer for the kind and positive comments. We have revised the manuscript, taking into account the Reviewer’s concerns below.

COMMENT 9. Some sentences could be restructured for better clarity and readability. For instance, in the sentence “For the specific microRNA let7a the assays were designed to cover the greatest possible number of CpG sites within the promoter region but considering length of the PCR amplicon, length of the target sequence and primers that avoided CpGs”, the phrase “but considering” is somewhat ambiguous in this context, and it is not entirely clear how it relates to the rest of the sentence.

ANSWER 9. We thank the Reviewer for the note.  The sentence was rephrased (lines 358-360).

COMMENT 10The authors may consider ensuring consistency in the use of terminology throughout the manuscript. For example, since “cigarette smoke” is abbreviated as “CS”, this abbreviation should be used consistently throughout the manuscript.

ANSWER 10. We thank the Reviewer for the noteWe have been used the CS abbreviation as suggested.

COMMENT 11.  While the grammar appears to be generally good, there are minor errors that could be corrected. For instance, the authors may wish to ensure correct use of articles (a, an, the) and prepositions (of, in, on). Also, they could check for correct punctuation, particularly in complex sentences.

ANSWER 11. We thank the Reviewer for the note.  We have been revised the manuscript as suggested.

Round 2

Reviewer 2 Report (New Reviewer)

No more questions.

Author Response

We thank the reviewer for the positive comment.

Reviewer 4 Report (New Reviewer)

Here are my few comments that needs to be corrected:

1) In line 49, comma is missing after cigarette smoke 

2) Line 83 begins with a case error for let-7a

3) There is typo for KRAS in line 105 in parenthesis

4) There is a sentence formation error in line 290-292. Needs to be re-written

Author Response

Dear Editors,

We would like to thank you for considering the manuscript entitled “Let-7a downregulation accompanied by KRAS mutation is predictive of lung cancer onset in Cigarette Smoke exposed mice” by Pulliero A. et al., and for sharing the Reviewers’ comments that certainly helped in improving the quality of the manuscript (ID: ijerph-2445830). We appreciated the Reviewers’ comments, and we revised the manuscript accordingly. Please find enclosed to the submission of the revised version of the manuscript the point-by point reply to the Reviewers’ comments. For clarity’s sake, changes in the revised MS are wrote in green color. 

1) In line 49, comma is missing after cigarette smoke . It has been added.

2) Line 83 begins with a case error for let-7°. It has been corrected.

3) There is typo for KRAS in line 105 in parenthesis. It has been corrected.

4) There is a sentence formation error in line 290-292. Needs to be re-written. It has been re-written.

This manuscript is a resubmission of an earlier submission. The following is a list of the peer review reports and author responses from that submission.

Round 1

Reviewer 1 Report

  1. Lack of context: The introduction lacks context about the significance and urgency of the topic. It only mentions the high fatality rate of lung cancer and its threat to public health without explaining the broader implications or the need for further research.

  2. Limited scope: The introduction focuses on the molecular and genetic aspects of lung cancer, specifically the role of let-7a and KRAS mutations. While these are important areas of research, the introduction could benefit from discussing other factors such as environmental, lifestyle, and socioeconomic factors that contribute to lung cancer.

  3. Technical language: The introduction contains technical language and acronyms that may be difficult for non-experts to understand, which could limit the accessibility of the research to a wider audience.

  4. Lack of clarity: Some sentences are not clear in their meaning or structure, making it difficult to understand the intended message. For example, sentence 65 is unclear in its phrasing and could benefit from rewording.

  5. Lack of citations: Some statements in the introduction lack citations, making it difficult for readers to verify the accuracy and reliability of the information. For example, sentence 4 mentions that lung cancer has the fastest increase in morbidity and mortality, but there is no citation to support this claim.

  6. The downregulation of let-7a expression by cigarette smoke was not significant after 2 weeks of exposure, which suggests that the effect may require longer periods of exposure to become apparent. Therefore, the short-term effects of cigarette smoke on let-7a expression may be underestimated.

  7. Although there was a trend towards decreasing methylation rate in the let-7a promoter with cigarette smoke exposure, no significant differences were detected. This may be due to the small sample size or the sensitivity of the assay used.

  8. The study found a significant increase in KRAS mutations in mice exposed to cigarette smoke. However, the study did not examine other potential genetic mutations or epigenetic changes that may contribute to lung cancer development.

  9. The sample size for the cancer incidence analysis was relatively small, which may limit the generalizability of the findings.

  10. No significant difference was detected when comparing cancer-bearing and cancer-free mice in terms of KRAS mutations, let-7a expression, and let-7a promoter methylation. This suggests that these biomarkers may not be useful for predicting lung cancer development in mice. However, the authors did not explore other potential biomarkers that may be associated with lung cancer development.

The Materials and Methods section has several drawbacks, including:

  1. Lack of detail: The section does not provide sufficient details about the methods and techniques used. For example, the section does not describe the equipment and software used for computational analysis.

  2. Missing information: The section lacks important information, such as the age and sex of the mice, which could affect the study results.

  3. Incomplete information: The section does not provide complete information on the experimental design, such as the sample size and the number of replicates used.

  4. Lack of justification: The section does not provide sufficient justification for the methods and techniques used, such as why a particular DNA extraction kit was chosen.

  5. Lack of reproducibility: The section does not provide sufficient information to replicate the study, which is essential for the reproducibility of the study.

Moderate editing of English language

Reviewer 2 Report

In this study, authors expose a very exciting objective: decipher interactions between genetics (KRAS mutations), epigenetics (Let-7a methylation and expression level), and environmental factors (cigarette smoke exposure) for the occurrence of lung cancer in a mouse model.

Unfortunately, both methods and results are not clearly exposed and fail to convince that "Let-7a downregulation accompanied by KRAS mutation is predictive of lung cancer onset in Cigarette Smoke exposed mice". As I could understand it, the main conclusion could be "Regardless of cigarette smoke exposure the correlation between Let-7a expression level and number of KRAS transversion at codons 12 and 13 is opposite between cancer-free and cancer-carriers of a spontaneous lung cancer mice model".

I report below only some of the main issues.

In the introduction, I did not understand the sentence "In lung adenocarcinomas, it was discovered that let-7a-3 hypomethylation increased the expression of miRNA and decreased the development of cancerous cells.” What is the “development of cancerous cells” if it is studied in an already cancerous tissue such as lung adenocarcinoma? The cited reference (n°10) does not study lung carcinoma, but it seems references numbers have been mixed.

About Let-7a expression:

In figure 1A, what is the unit of y axis? According to the method section (line 301), it is a fold change, but compared to what? Are “CS short” and “CS long” group compared to the same mice or to age-stratified mice?

About Let-7a promoter methylation:

The sentence “Accordingly, a not significant trend towards a decrease in methylation rate going on with CS exposure was observed” (line 126) is not necessary as it just repeats the previous one and encourages the reader to draw inappropriate conclusions. There is almost no change in methylation rate between the three groups. A decrease in methylation with cigarette smoke would anyway not explain a decrease in Let-7a expression level.

Was the methylation analyzed in blood or in lung? In the “method” section DNA was extracted from blood, but reported KRAS mutations actually occurred in the lung.

About KRAS mutation:

 “The percentage of G/T transversions in codons 12 and 13 was 2.3±0.40 % in sham” (line 143) : mutation compared to which reference if “sham” is the control group??

In Figure 3B (line 152): The sentence “KRAS mutation was significantly higher in CS exposed mice as compared to sham” is unclear. It should be “the number of KRAS mutations was significantly higher…”.  What does this figure exactly show? Is it the total number of tranversions in each group or the percentage of mice with at least one transversion? In any case, what do the error bars mean?

All mutations (including transitions) should be listed to show if they are pathogenic or not, and if the number of transversion in different from the number of transitions.

About cancer incidence:

Line 154: 4% of unexposed mice develop lung cancer? Are they wild type mice? Two weeks of smoke exposure prevents lung cancer occurrence?

About comparison between cancer-bearing and cancer-free mice:

Line 182: “The only biomarker distinguishing between cancer free and cancer bearing mice was the combination of let-7a downregulation intensity together with KRAS mutation amount.” This sentence is confusing. It is not a biomarker: the relation between those two parameters was different between normal lung and cancer. Is this relations dependent of exposition to smoke?

More generally, the authors never mention the number of mice in each group (!), nor the precise experimental design. How do mice were exposed to cancer smoke? What is “sham exposition”? A table with age, smoke exposition, cancer type and genetic/epigenetic results for each mouse would be much more useful than most of the figures.